# REINFORCEMENT LEARNING WITH LATENT FLOW

## ABSTRACT

Temporal information is essential to learning effective policies with Reinforcement Learning (RL). However, current state-of-the-art RL algorithms either assume that such information is given as part of the state space or, when learning from pixels, use the simple heuristic of frame-stacking to implicitly capture temporal information present in the image observations. This heuristic is in contrast to the current paradigm in video classification architectures, which utilize explicit encodings of temporal information through methods such as optical flow and two-stream architectures to achieve state-of-the-art performance. Inspired by leading video classification architectures, we introduce the **F**low of **La**tents for **Re**inforcement Learning (*Flare*), a network architecture for RL that explicitly encodes temporal information through latent vector differences. We show that Flare (i) recovers optimal performance in state-based RL without explicit access to the state velocity, solely with positional state information, (ii) achieves state-of-the-art performance on pixel-based continuous control tasks within the DeepMind control benchmark suite, (iii) is the most sample efficient model-free pixel-based RL algorithm on challenging environments in the DeepMind control suite such as quadruped walk, hopper hop, finger turn hard, pendulum swing, and walker run, outperforming the prior model-free state-of-the-art by **1.9**× and **1.5**× on the 500k and 1M step benchmarks, respectively, and (iv), when augmented over rainbow DQN, outperforms or matches the baseline on a diversity of challenging Atari games at 50M time step benchmark.

## 1    INTRODUCTION

Reinforcement learning (RL) (Sutton & Barto, 1998) holds the promise of enabling artificial agents to solve a diverse set of tasks in uncertain and unstructured environments. Recent developments in RL with deep neural networks have led to tremendous advances in autonomous decision making. Notable examples include classical board games (Silver et al., 2016; 2017), video games (Mnih et al., 2015; Berner et al., 2019; Vinyals et al., 2019), and continuous control (Schulman et al., 2017; Lillicrap et al., 2016; Rajeswaran et al., 2018). A large body of research has focused on the case where an RL agent is equipped with a compact state representation. Such compact state representations are typically available in simulation (Todorov et al., 2012; Tassa et al., 2018) or in laboratories equipped with elaborate motion capture systems (OpenAI et al., 2018; Zhu et al., 2019; Lowrey et al., 2018). However, state representations are seldom available in unstructured real-world settings like the home. For RL agents to be truly autonomous and widely applicable, sample efficiency and the ability to act using raw sensory observations like pixels is crucial. Motivated by this understanding, we study the problem of efficient and effective deep RL from pixels.

A number of recent works have made progress towards closing the sample-efficiency and performance gap between deep RL from states and pixels (Laskin et al., 2020b;a; Hafner et al., 2019a; Kostrikov et al., 2020). An important component in this endeavor has been the extraction of high quality visual features during the RL process. Laskin et al. (2020a) and Stooke et al. (2020) have shown that features learned either explicitly with auxiliary losses (reconstruction or contrastive losses) or implicitly (through data augmentation) are sufficiently informative to recover the agent's pose information. While existing methods can encode positional information from images, there has been little attention devoted to extracting temporal information from a stream of images. As a result, existing deep RL methods from pixels struggle to learn effective policies on more challenging continuous control environments that deal with partial observability, sparse rewards, or those that require precise manipulation.

Current approaches in deep RL for learning temporal features are largely heuristic in nature. A commonly employed approach is to stack the most recent frames as inputs to a convolutional neural network (CNN). This can be viewed as a form of early fusion (Karpathy et al., 2014), where information from the recent time window is combined immediately at the pixel level for input to the CNN. In contrast, modern video recognition systems use alternate architectures that employ optical flow and late fusion (Simonyan & Zisserman, 2014), where frames are processed individually with CNN layers before fusion and downstream processing. Such a late fusion approach is typically beneficial due to better performance, fewer parameters, and the ability to use multi-modal data (Jain et al., 2019; Chebotar et al., 2017). However, it is not straightforward how to port such architectures to RL. Comput-

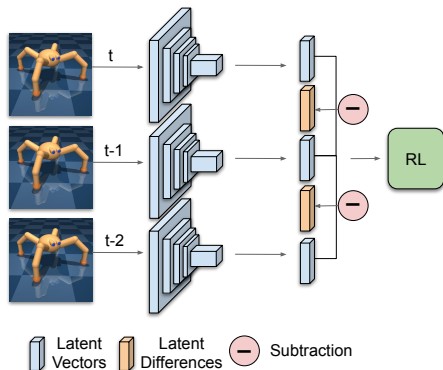

Figure 1: **F**low of **La**tents for **Re**inforcement Learning (*Flare*) architecture. Input frames are first encoded individually by the same encoder. The resulting latent vectors are then concatenated with their latent differences before being passed to the downstream RL algorithm.

ing optical flow in real-time for action selection can be computationally infeasible in applications with fast control loops like robotics. In our experiments, we also find that a naive late fusion architecture minus the optical flow yields poor results in RL settings (see Section 6.3). This observation is consistent with recent findings in related domains like visual navigation (Walsman et al., 2019).

To overcome the above challenges, we develop **F**low of **La**tents for **Re**inforcement Learning (*Flare*), a new architecture for deep RL from pixels (Figure 1). Flare can be interpreted as a *structured late fusion* architecture. Flare processes each frame individually to compute latent vectors, similar to a standard late fusion approach (see Figure 1). Subsequently, temporal differences between the latent feature vectors are computed and fused along with the latent vectors by concatenation for downstream processing. By incorporating this structure of temporal difference in latent feature space, we provide the learning agent with appropriate inductive bias. In experiments, we show that Flare (i) recovers optimal performance in state-based RL without explicit access to the state velocity, solely with positional state information, (ii) achieves state-of-the-art performance compared to model-free methods on several challenging pixel-based continuous control tasks within the DeepMind control benchmark suite, namely Quadruped Walk, Hopper Hop, Finger Turn-hard, Pendulum Swingup, and Walker Run, and (iii) is the most sample efficient model-free pixel-based RL algorithm across these tasks, outperforming the prior model-free state-of-the-art RAD by **1.9**× and **1.5**× on the 500k and 1M environment step benchmarks, respectively.

## 2 RELATED WORK

**Pixel-Based RL** The ability of an agent to autonomously learn control policies from visual inputs can greatly expand the applicability of deep RL (Dosovitskiy et al., 2017; Savva et al., 2019). Prior works have used CNNs to extend RL algorithms like PPO (Schulman et al., 2017), SAC (Haarnoja et al., 2018), and Rainbow (Hessel et al., 2017) to pixel-based tasks. Such direct extensions have typically required substantially larger number of environment interactions when compared to the state-based environments. In order to improve sample efficiency, recent efforts have studied the use of auxiliary tasks and loss functions (Yarats et al., 2019; Laskin et al., 2020b; Schwarzer et al., 2020), data augmentation (Laskin et al., 2020a; Kostrikov et al., 2020), and latent space dynamics modeling (Hafner et al., 2019b;a). Despite these advances, there is still a large gap between the learning efficiency in state-based and pixel-based environments in a number of challenging benchmark tasks. Our goal in this work is to identify where and how to improve pixel-based performance on this set of challenging control environments.

**Neural Network Architectures in RL** The work of Mnih et al. (2015) combined Q-learning with CNNs to achieve human level performance in Atari games. In this work, Mnih et al. (2015) concatenate the most recent 4 frames and use a convolutional neural network to output the Q values. In 2016, Mnih et al. (2016) proposed to use a shared CNN among frames to extract visual features and aggregate the temporal information with LSTM. The same architectures have been adopted by most

works till date (Laskin et al., 2020b; Schwarzer et al., 2020; Kostrikov et al., 2020; Laskin et al., 2020a). The development of new architectures to better capture temporal information in a stream of images has received little attention in deep RL, and our work aims to fill this void. Perhaps closest to our motivation is the work of Amiranashvili et al. (2018) who explicitly use optical flow as an extra input to the RL policy. However, this approach requires additional information and supervision signal to train the flow estimator, which could be unavailable or inaccurate in practice. In contrast, our approach is a simple modification to existing deep RL architectures and does not require any additional auxiliary tasks or supervision signals.

**Two-Stream Video Classification** In video classification tasks, such as activity recognition (Soomro et al., 2012), there are a large body of works on how to utilize temporal information (Donahue et al., 2015; Ji et al., 2012; Tran et al., 2015; Carreira & Zisserman, 2017; Wang et al., 2018; Feichtenhofer et al., 2019). Of particular relevance is the two-stream architecture of Simonyan & Zisserman (2014), where one CNN stream takes the usual RGB frames, while the other the optical flow computed from the RGB values. The features from both streams are then late-fused to predict the activity class. Simonyan & Zisserman (2014) found that the two-stream architecture yielded a significant performance gain compared to the single RGB stream counterpart, indicating the explicit temporal information carried by the flow plays an essential role in video understanding. Instead of directly computing the optical flow, we propose to capture the motion information in latent space to avoid computational overheads and potential flow approximation errors. Our approach also could focus on domain-specific motions that might be overlooked in a generic optical flow representation.

## 3 BACKGROUND

**Soft Actor Critic** (SAC) (Haarnoja et al., 2018) is an off-policy actor-critic RL algorithm for continuous control with an entropy maximization term augmented to its score function to encourage exploration. SAC learns a policy network $\pi_\psi(a_t|\mathbf{o}_t)$ and critic networks $Q_{\phi_1}(\mathbf{o}_t, a_t)$ and $Q_{\phi_2}(\mathbf{o}_t, a_t)$ to estimate state-action values. The critic $Q_{\phi_i}(\mathbf{o}_t, a_t)$ is optimized to minimize the (soft) Bellman residual error:

$$\mathcal{L}_Q(\phi_i) = \mathbb{E}_{\tau \sim \mathcal{B}} \left[ \left( Q_{\phi_i}(\mathbf{o}_t, a_t) - (r_t + \gamma V(\mathbf{o}_{t+1})) \right)^2 \right], \tag{1}$$

where $r$ is the reward, $\gamma$ the discount factor, $\tau = (\mathbf{o}_t, a_t, \mathbf{o}_{t+1}, r_t)$ is a transition sampled from replay buffer $\mathcal{B}$, and $V(\mathbf{o}_{t+1})$ is the (soft) target value estimated by:

$$V(\mathbf{o}_{t+1}) = \left( \min_i Q_{\bar{\phi}_i}(\mathbf{o}_{t+1}, a_{t+1}) - \alpha \log \pi_\psi(a_{t+1}|\mathbf{o}_{t+1}) \right), \tag{2}$$

where $\alpha$ is the entropy maximization coefficient. For stability, in eq. 2, $Q_{\bar{\phi}_i}$ is the exponential moving average of $Q_{\phi_i}$'s over training iterations. The policy $\pi_\psi$ is trained to maximize the expected return estimated by $Q$ together with the entropy term

$$L_\pi(\psi) = -\mathbb{E}_{a_t \sim \pi} \left[ \min_i Q_{\phi_i}(\mathbf{o}_t, a_t) - \alpha \log \pi_\psi(a_t|\mathbf{o}_t) \right], \tag{3}$$

where $\alpha$ is also a learnable parameter.

**Reinforcement Learning with Augmented Data** (RAD) (Laskin et al., 2020a) is a recently proposed training technique. In short, RAD pre-processes raw pixel observations by applying random data augmentations, such as random translation and cropping, for RL training. As simple as it is, RAD has taken many existing RL algorithms, including SAC, to the next level. For example, on many DMControl (Tassa et al., 2018) benchmarks, while vanilla pixel-based SAC performs poorly, RAD-SAC—i.e. applying data augmentation to pixel-based SAC—achieves state-of-the-art results both in sample efficiency and final performance. In this work, we refer RAD to RAD-SAC and the augmentation used is random translation.

**Rainbow DQN** is an extension of the Nature Deep Q Network (DQN) (Mnih et al., 2015), which combines multiple follow-up improvements of DQN to a single algorithm (Hessel et al., 2017). In summary, DQN (Mnih et al., 2015) is an off-policy RL algorithm that leverages deep neural networks (DNN) to estimate the Q value directly from the pixel space. The follow-up works Rainbow DQN bring together to enhance the original DQN include double Q learning (Hasselt, 2010), prioritized experience replay (Schaul et al., 2015), dueling network (Wang et al., 2016), noisy network (Fortunato et al., 2017), distributional RL (Bellemare et al., 2017) and multi-step returns (Sutton & Barto, 1998).

Rainbow DQN is one of the state-of-the-art RL algorithms on the Atari 2600 benchmark (Bellemare et al., 2013). We thus adopt an official implementation of Rainbow (Quan & Ostrovski, 2020) as our baseline to directly augment Flare on top.

## 4 MOTIVATION

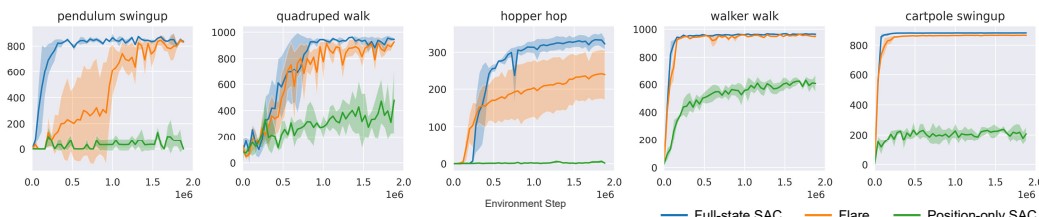

Figure 2: (i) full-state SAC (blue) where input contains both pose and temporal information; (ii) position-only SAC (green) with only pose information as input; (iii) Flare applied to the state space (orange) with pose information and velocity approximations through pose offsets as input. While full-state SAC efficiently learns the optimal policy, position-only SAC recovers suboptimal policies and fails learning at all in some cases. Meanwhile, the fusion of approximated velocities in Flare is able to recover the optimal policy nearly as efficiently as the full state SAC in most cases. Results are averaged over 3 seeds with standard deviation.

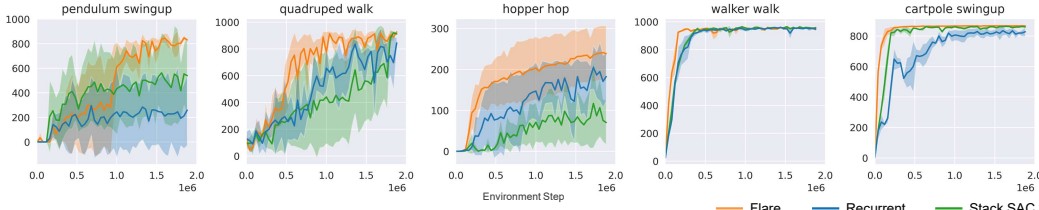

Figure 3: We compare Flare to 2 SAC variants: i) Stack SAC (green) receives consecutive positional states $(s_t, s_{t-1}, s_{t-2}, s_{t-3})$ as input, whereas positional-only SAC receives $(s_t)$ and Flare receives $(s_t, \delta_t)$ where $\delta_t = (s_t - s_{t-1}, s_{t-1} - s_{t-2}, s_{t-2} - s_{t-3})$. ii) Recurrent SAC (blue) uses recurrent layers to process a series of states. Despite of the implicit access to temporal information between consecutive states, Stack SAC and Recurrent perform significantly worse than Flare on most environments, highlighting the benefit of explicit fusion of temporal information. Results are averaged over three seeds.

We motivate our method by investigating the importance of temporal information in state-based RL. Our investigation utilizes five diverse DMControl (Tassa et al., 2018) tasks. The full state for these environments includes both the agent's pose information, such as the joints' positions and angles, as well as temporal information, such as the joints' translational and angular velocities. We train two variants with SAC—one variant where the agent receives the full state as input (full-state SAC), and the other with the temporal information masked out, i.e. the agent only receives the pose information as its input (position-only SAC). The resulting learning curves are in Figure 2. While the full-state SAC learns the optimal policy quickly, the position-only SAC learns much sub-optimal policies, which often fail entirely. It is therefore clearly shown that effective policies cannot be learned from positional information alone, and that temporal information is crucial for efficient learning.

While full-state SAC can receive velocity information from internal sensors in simulation, in the more general case such as learning from pixels, such information is often not readily available. For this reason, we investigate whether we can explicitly approximate temporal information as the difference between two consecutive states. If the input is the positional state, then this positional difference roughly approximates the agent's velocity. Given poses $s_t^p, s_{t-1}^p, s_{t-2}^p, s_{t-3}^p$ at time $t, t-1, t-2, t-3$, we compute the positional offset $\delta_t = (s_t - s_{t-1}, s_{t-1} - s_{t-2}, s_{t-2} - s_{t-3})$, and provide the fused vector $(s_t, \delta_t)$ to the SAC agent. This procedure precisely describes the state-based version of Flare. Results shown in Figure 2 demonstrate that state-based Flare significantly outperforms the position-only SAC. Furthermore, state-based Flare achieves optimal asymptotic performance, and its learning

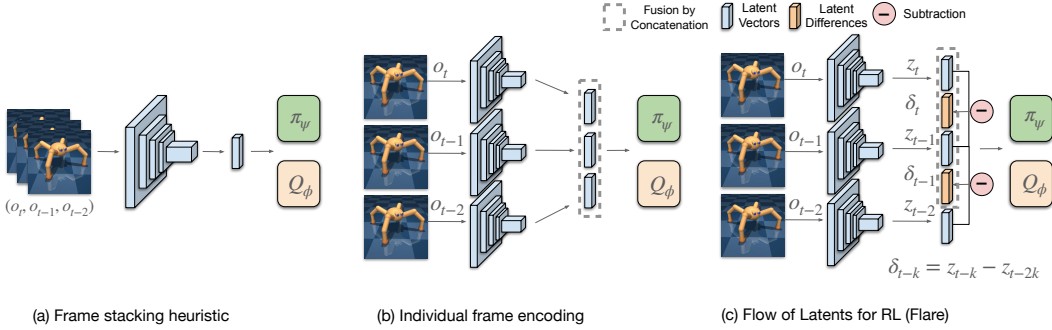

Figure 4: **F**low of **La**tents for **Re**inforcement Learning (*Flare*). In panel (a) we show the architecture for the frame stacking heuristic, in (b) we show an alternative to the frame stacking hueristic by encoding each image individually, and in (c) we show the Flare architecture which encodes images individually, computes the feature differences, and fuses the differences together with the latents.

efficiency is comparable to full-state SAC in most environments. Given that the position-only SAC (which utilizes $s_t$ alone) has only partial information compared to Flare that utilizes $(s_t, \delta_t)$, we also investigate a variant where we provide consecutive positions $(s_t, s_{t-1}, s_{t-2}, s_{t-3})$ to the SAC agent. We call this variant Stack SAC, since it is identical to the frame-stack heuristic used in pixel-based RL. Results in Figure 3 show that Flare still significantly outperforms the Stack SAC. It suggests that the well-structured inductive bias in the form of temporal-position fusion is essential for efficient learning.

A recurrent structure is an alternative approach to process temporal information. We implement an SAC variant with recurrent modules (Recurrent SAC) and compare it with Flare. Specifically, we pass a sequence of poses $s_t^p, s_{t-1}^p, s_{t-2}^p, s_{t-3}^p$ through an LSTM cell. The number of the LSTM hidden units $h$ is set to be the same as the dimension of $\delta_t$ in Flare. The trainable parameters of the LSTM cell are updated to minimize the critic loss. Recurrent SAC is more complex to implement and requires longer wall-clock training time, but performs worse than Flare as shown in Figure 3.

Our findings from the state experiments in Figure 2 and Figure 3 suggest that (i) temporal information is crucial to learning effective policies in RL and (ii) approximating temporal information in the absence of sensors that provide explicit measurements is sufficient in most cases. When learning from pixels, it is common to assume the absence of specialized sensors for reading out temporal information. We therefore hypothesize that explicit fusion of temporal information approximated directly from pixel-level inputs can improve the efficiency of learning control policies.

## 5 REINFORCEMENT LEARNING WITH LATENT FLOW

To date, frame stacking is the most common way of pre-processing pixel-based input to convey temporal information for RL algorithms. This heuristic, introduced by Mnih et al. (2015), has been largely untouched since its inception and is used in most state-of-the-art RL architectures. However, our observations from the experiments run on state input in Section 4 suggest an alternative to the frame stacking heuristic through the explicit inclusion of temporal information as part of the input. To learn effective control policies from pixels, we seek a general approach to explicitly incorporate temporal information that can be coupled to any base RL algorithm with minimal modification. To this end, we propose the **F**low of **La**tents for **Re**inforcement Learning (*Flare*) architecture. Our proposed method calculates differences between the latent encodings of individual frames and fuses the feature differences and latent embeddings before passing them as input to the base RL algorithm, as shown in Figure 4. We demonstrate Flare on top of 2 state-of-the-art model-free off-policy RL baselines, RAD-SAC (Laskin et al., 2020a) and Rainbow DQN (Hessel et al., 2017), though any RL algorithm can be used in principle.

| Task | Flare (500K) | RAD (500K) | Flare (1M) | RAD (1M) |
|---|---|---|---|---|
| Quadruped Walk | $\mathbf{296} \pm 139$ | $206 \pm 112$ | $\mathbf{488} \pm 221$ | $322 \pm 229$ |
| Pendulum Swingup | $\mathbf{242} \pm 152$ | $79 \pm 73$ | $\mathbf{809} \pm 31$ | $520 \pm 321$ |
| Hopper Hop | $\mathbf{90} \pm 55$ | $40 \pm 41$ | $\mathbf{217} \pm 59$ | $211 \pm 27$ |
| Finger Turn hard | $\mathbf{282} \pm 67$ | $137 \pm 98$ | $\mathbf{661} \pm 315$ | $249 \pm 98$ |
| Walker Run | $426 \pm 33$ | $\mathbf{547} \pm 48$ | $556 \pm 93$ | $\mathbf{628} \pm 39$ |

Table 1: Evaluation on 5 benchmark tasks around 500K and 1M environment steps. We evaluate over 5 seeds, each of 10 trajectories and show the mean $\pm$ standard deviation across runs.

| | Flare (50M) | Rainbow (50M) | | Flare (50M) | Rainbow (50M) |
|---|---|---|---|---|---|
| Assault | $9466 \pm 1928$ | $\mathbf{10123} \pm 2061$ | Breakout | $\mathbf{330} \pm 10$ | $321 \pm 34$ |
| Freeway | $\mathbf{34} \pm 0$ | $\mathbf{34} \pm 0$ | Krull | $8423 \pm 173$ | $\mathbf{8030} \pm 717$ |
| Montezuma | $\mathbf{400} \pm 0$ | $0 \pm 0$ | Seaquest | $\mathbf{8362} \pm 1180$ | $4521 \pm 3554$ |
| Up n Down | $\mathbf{44055} \pm 12746$ | $24568 \pm 2216$ | Tutankham | $\mathbf{240} \pm 7$ | $148 \pm 16$ |

Table 2: Evaluation on 8 benchmark Atari games at 50M training steps over 3 seeds.

## 5.1 LATENT FLOW

In computer vision, the most common approach to explicitly inject temporal information of a video sequence is to compute dense optical flow between consecutive frames (Simonyan & Zisserman, 2014). Then the RGB and the optical flow inputs are individually fed into two streams of encoders and the features from both streams are fused in the later stage of the network. However, two-stream architectures with optical flow are not directly applicable to RL. The main issue is that the computation of optical flow is slow: during inference, it is often prohibitively expensive to compute in real-time for applications with fast control loops like robotics; during training, optical flow calculation adds significant overhead to the wallclock training time in online learning settings like RL. While video architectures can utilize a memory bank, such that optical flow need only be pre-computed once for the entire dataset, RL training is done dynamically and on the fly, and computing optical flow at each step is therefore costly.

---

**Algorithm 1:** Pixel-based Flare Inference

Given $\pi_\psi, f_{\mathrm{CNN}}$;
**for** *each environment step $t$* **do**
  $z_j = f_{\mathrm{CNN}}(o_j), j = t-k, .., t$;
  $\delta_j = z_j - z_{j-1}, j = t-k+1, .., t$;
  $\mathbf{z}_t = (z_{t-k+1}, \cdots, z_t, \delta_{t-k+1}, \cdots, \delta_t)$;
  $a_t \sim \pi_\psi(a_t | \mathbf{z}_t)$;
  $o_{t+1} \sim p(o_{t+1} | a_t, \mathbf{o}_t = (o_t, o_{t-1}..o_{t-k}))$;
**end**

---

To address this challenge and motivated by experiments in Section 4, we propose an alternative architecture that is similar in spirit to the two-stream networks for video classification. Rather than computing optical flow directly, we approximate temporal information in the latent space. Instead of encoding a stack of frames at once, we use a frame-wise CNN to encode each individual frame. Then we compute the differences between the latent encodings of consecutive frames, which we refer to as *latent flow*. Finally, the latent features and the latent flow are fused together through concatenation before getting passed to the downstream RL algorithm. We call the proposed architecture as **F**low of **La**tents for **Re**inforcement Learning (*Flare*).

While Flare is a broadly applicable technique, for clarity of exposition, we select RAD as the base algorithm to elaborate the execution of Flare. We also use RAD later on in our experiments as the comparative baseline (Section 6). The RAD architecture, shown in Figure 4a, stacks multiple data augmented frames observed in the pixel space and encodes them altogether through an CNN. This can be viewed as a form of early fusion (Karpathy et al., 2014). Another preprocessing option is to encode each frame individually through a shared frame-wise encoder and perform late fusion of the resulting latent features, as shown in Figure 4b. However, we find that simply concatenating the latent features results in inferior performance when compared to the frame stacking heuristic, which we further elaborate in Section 6.3. We conjecture that pixel-level frame stacking benefits from leveraging both the CNN and the fully connected layers to process temporal information, whereas latent-level stacking does not propagate temporal information back through the CNN encoder. Based on this conjecture, we explicitly compute the latent flow $\delta_t = z_t - z_{t-1}$ while detaching the $z_{t-1}$ gradients when computing $\delta_t$. We fuse the latent flow $\delta_t$ with the latent embedding $z_t$, and pass the fused input to the actor and critic networks as shown in Figure 4c. We provide pseudocode that illustrates how to do inference with Flare in Algorithm 1; during training, the encodings of latent features and latent flow are done in the same way except with augmented observations.

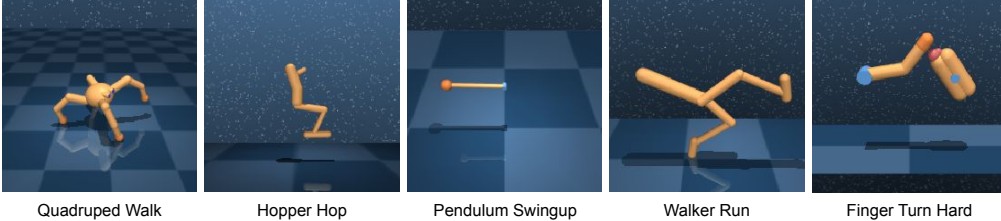

Quadruped Walk Hopper Hop Pendulum Swingup Walker Run Finger Turn Hard

Figure 5: We choose the following environments for our main experiments – (i) quadruped walk, which requires coordination of multiple joints, (ii) hopper hop, which requires hopping while maintaining balance, (iii) pendulum swingup, an environment with sparse rewards, (iv) walker run, which requires the agent to maintain balance at high speeds, and (v) finger turn hard, which requires precise manipulation of a rotating object. These environments are deemed challenging because prior state-of-the-art model-free pixel-based methods (Laskin et al., 2020b; Kostrikov et al., 2020; Laskin et al., 2020a) either fail to reach the asymptotic performance of state SAC or learn less efficiently.

# 6 EXPERIMENTS

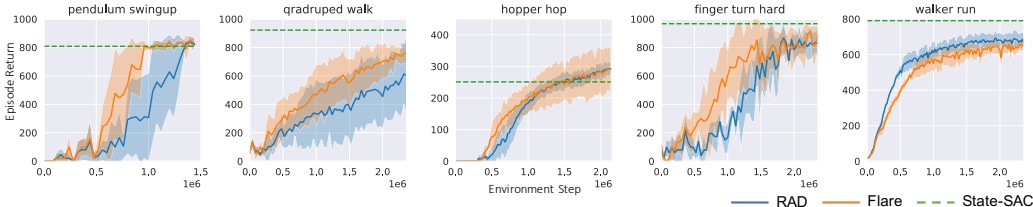

Figure 6: We compare the performance of Flare to RAD, a state-of-the-art algorithm and the base algorithm used in Flare, on five challenging environments. Pendulum Swingup are trained over $1.5e6$ and the rest $2.5e6$. We see that Flare substantially outperforms RAD on a majority (3 out of the 5) of environments, while being competitive in the remaining. While not closing the gap between pixel and state-based performance entirely, Flare is closer to state-based performance than prior methods, and is the state-of-the-art pixel-based model-free algorithm on most of these challenging environments. Results are averaged over 5 random seeds with standard deviation (shaded regions).

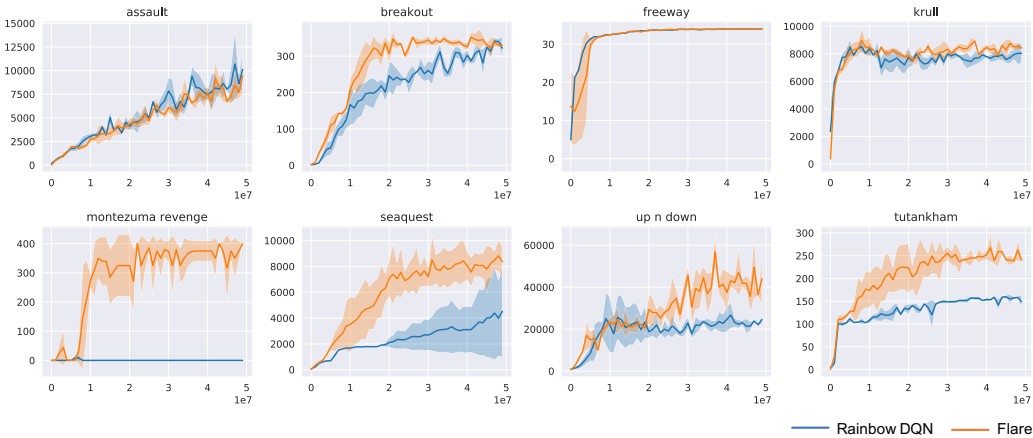

Figure 7: We compare Rainbow DQN and Flare on 8 Atari games over 50M training steps. Flare substantially enhances a majority (5 out of 8) of the games over the baseline Rainbow DQN while matching the rest. Results are averaged over 3 random seeds with standard deviation (shaded regions).

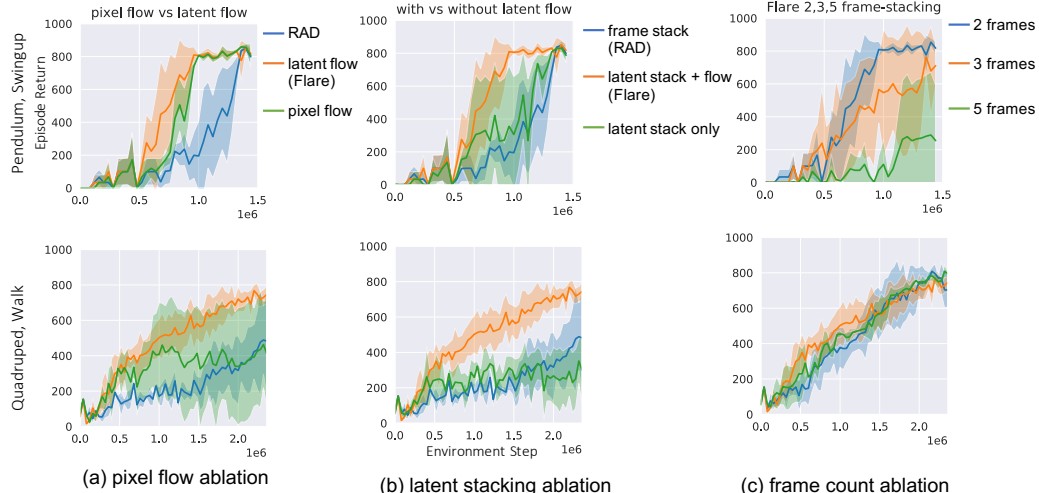

Figure 8: We perform three ablation studies. (a) *pixel flow ablation*: we compare Flare to a variant where the differences are computed directly in pixel space (pixel flow) and find that latent flow is more stable and achieves better performance. (b) *Latent stack ablation*: in this experiment, we fuse the latent vectors without the temporal approximation. We find that this method performs significantly worse than Flare, and on quadruped fails entirely, suggesting that fusing explicit temporal information is crucial. (c) *Frames count ablation*: We test whether adding more frames increases performance for Flare. We find that including additional input frames either does not change or degrades performance.

We first introduce the 5 core challenging continuous control tasks from DMControl suite (Tassa et al., 2018) that our experiments focus on. Next we present the main experimental results, where we show that Flare achieves substantial performance gains over the base algorithm RAD (Laskin et al., 2020a). Finally, we conduct a series of ablation studies to stress test the design choices of the Flare architecture.

## 6.1 ENVIRONMENTS AND EVALUATION METRICS

**The DeepMind Control Suite** (DMControl) (Tassa et al., 2018), based on MuJoCo (Todorov et al., 2012), is a commonly used benchmark for continuous control from pixels. Prior works such as DrQ (Kostrikov et al., 2020) and RAD (Laskin et al., 2020a) have made substantial progress on this benchmark and closed the gap between state-based and pixel-based efficiency on the simpler environments in the suite, such as Reacher Easy, Ball-in-cup Catch, Finger Spin, Walker Walk, Cheetah Run, Cartpole Swingup. However, current pixel-based RL algorithms struggle to learn optimal policies efficiently in more challenging environments that feature partial observability, sparse rewards, or precise manipulation. In this work, we study more challenging tasks from the suite to better showcase the efficacy of our proposed method. The 5 environments, listed in Figure 5, include Walker Run (requires maintaining balance with speed), Quadruped Walk (partially observable agent morphology), Hopper Hop (locomotion with sparse rewards), Finger Turn-hard (precise manipulation), and Pendulum Swingup (torque control with sparse rewards). For evaluation, we benchmark performance at 500K and 1M *environment steps* and compare against RAD.

**The Atari 2600 Games** (Bellemare et al., 2013) is another highly popular RL benchmark. Recent efforts have let to a range of highly successful algorithms (Espeholt et al., 2018; Hessel et al., 2017; Kapturowski et al., 2018; Hafner et al., 2019a; Badia et al., 2020) to solve Atari games directly from pixel space. A representative state-of-the-art is Rainbow DQN (see Section 3). We adopt the official Rainbow DQN implementation (Quan & Ostrovski, 2020) as our baseline. Then we simply modify the model architecture to incorporate Flare while retaining all the other default settings, including hyperparameters and preprocessing. To ensure comparable model capacity, the Flare network halves the number of convolutional channels and adds a bottleneck FC layer to reduce latent dimension before entering the Q head (code in the Supplementary Materials). We evaluate on a diverse subset of Atari games at 50M *training steps*, namely Assault, Breakout, Freeway, Krull, Montezuma Revenge, Seaquest, Up n Down and Tutankham, to assess the effectiveness of Flare.

## 6.2 Main Results

**DMControl:** Our main experimental results on the five DMControl tasks are presented in Figure 6 and Table 1. We find that Flare outperforms RAD in terms of both final performance and sample efficiency for majority (3 out of 5) of the environments, while being competitive on the remaining environments. Specifically, Flare attains similar asymptotic performance to state-based RL on Pendulum Swingup, Hopper Hop, and Finger Turn-hard. For Quadruped Walk, a particularly challenging environment due to its large action space and partial observability, Flare learns much more efficiently than RAD and achieves a higher final score. Moreover, Flare outperforms RAD in terms of sample efficiency on all of the core tasks except for Walker Run as shown in Figure 6. The 500k and 1M environment step evaluations in Table 1 show that, on average, Flare achieves **1.9**× and **1.5**× higher scores than RAD at the 500k step and the 1M step benchmarks, respectively. Though our investigation primarily focuses on these 5 challenging environments, we also show in Appendix A.1 that Flare matches the state-of-the-art on the 6 simpler environments.

**Atari:** The results on the 8 Atari games are in Figure 7 and Table 3. Again, we observe substantial performance gain from Flare on the majority of the games while being equally competitive to the baseline Rainbow DQN on the remaining games. In Appendix A.2, we also show that Flare performs competitively when comparing against other DQN variants at 50M training steps.

## 6.3 Ablation Studies

We ablate a number of components of the Flare architecture on the Quadruped Walk and Pendulum Swingup environments to stress test the Flare architecture. The results shown in Figure 8 aim to answer the following questions:

**Q1**: *Do we need latent flow or is computing pixel differences sufficient?* While Flare proposes a late fusion of latent differences with the latent embeddings, a simpler approach is an early fusion of pixel differences with the pixel input, which we call pixel flow. We compare Flare to pixel flow in Figure 8 (left) and find that, while pixel flow outperforms RAD, it is significantly less efficient and less stable than Flare, particularly on Quadruped Walk. This ablation suggests that late fusion temporal information after encoding the image is preferable to early fusion.

**Q2**: *Are the gains coming from latent flow or individual frame-wise encoding?* Next, we address the potential concern that the performance gain of Flare stems from the frame-wise ConvNet architectural modification instead of the fusion of latent flow. Concretely, we follow the exact architecture and training as Flare, but instead of concatenating the latent flow, we concatenate each frame's latent after the convolution encoders directly as described in Figure 4 (b). This ablation is similar in spirit to the state-based experiments in Figure 3. The learning curves in Figure 8 (center) show that individual frame-wise encoding is not the source of increased performance. While on par with RAD on Pendulum Swingup, on Quadruped Walk frame-wise encoding performs worse. Flare's improved performance over RAD is therefore most likely a result of the explicit fusion of latent flow.

**Q3**: *How does the input frame count affect performance?* Lastly, we compare stacking 2, 3, and 5 frames in Flare in Figure 8 (right). We find that changing the number of stacked frames does not significantly impact the locomotion task, quadruped walk, but Pendulum Swingup tends to be more sensitive to this hyperparameter. Interestingly, the optimal number of frames for Pendulum Swingup is 2, and more frames can in fact degrade Flare's performance, indicating that the immediate position and velocity information is the most critical to learn effective policies on this task. We hypothesize that Flare trains more slowly with increased frame count on Pendulum Swingup due to the presence of unnecessary information that the actor and critic networks need to learn to ignore.

## 7 Conclusion

We propose Flare, an architecture for RL that explicitly encode temporal information by computing flow in the latent space. In experiments, we show that in the state space, Flare can recover the optimal performance with only state positions and no access to the state velocities. In the pixel space, Flare improves upon the state-of-the-art model-free RL algorithms on the majority of selected tasks in the DMControl and Atari suites, while matching in the remaining. Integrating Flare with model-based RL is a potential direction for future works.

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
