# OpenReview forum: "Reinforcement Learning with Latent Flow"
_ICLR.cc/2021/Conference — Reject_

### Official Review · AnonReviewer3 · 2020-10-28
**Official Blind Review #3**

**Rating:** 7
**Confidence:** 4

**Review:**

Summary:
This work presents a simple technique (Flare) to incorporate explicit temporal information to enable effective RL policy learning in challenging continuous control environments using pixel-based state representations. The approach is inspired from recent advances in the video recognition approaches which employ optical flow information and late fusion to incorporate temporal information. Typically, RL algorithms employ a frame-stacking heuristic to incorporate temporal information (early fusion). Though computing optical flow is slow and can been prohibitive for real-time applications, the authors present a simple alternative to using optical flow, i.e. difference of latent state vectors as a proxy for explicitly encoding motion information with latent vectors representing the observed state (late fusion). Experimental results on challenging continuous control tasks in the DMControl Suite show that Flare can achieve up to 1.9x higher scores than a baseline algorithm (RAD) which uses the frame-stacking heuristic to incorporate temporal information.

########################

Pros:
- The presented approach provides an effective alternative to the frame stacking heuristic for incorporating temporal information in RL with pixel-based state representations. The presented methodology (concatenation of latent state vector differences) thus learns effective policies on challenging continuous control environments.
- The presented approach can easily modify any RL algorithm operating on pixel-based state representations to encode temporal information.
- The paper is well-written, clear and easy to follow.
- The idea is well-motivated with experiments in environments with low-dimensional state spaces. The results from the motivation section show the importance of temporal information, and specifically explicit temporal information to learn effective policies.
- Ablation study clearly highlights the merits of including explicit temporal information with late fusion ruling out other techniques like pixel-based flow (early fusion) and the effect of independent convolutional feature extraction for different image frames.

########################

Cons:
- Comparisons to other approaches mentioned in Section 2 are missing.  For example, how would Flare compare in performance and sample efficiency to LSTM based RL methods (such as those under the “neural network architectures” subsection listed under Section 2)?
- Flare does not outperform RAD in all environments (such as hopper hop and walker run). It is unclear why Flare works well on some environments and not so on others. In fact, Flare performs worse than RAD on walker run. Why so?


########################

Reason for score:
The approach is well-motivated, and the paper is clearly written. The experiments comparing with an early fusion approach (RAD) and related ablative analysis highlighting that explicit late fusion of temporal information is key to improved performance are well-done and prove the effectiveness of the Flare. However, comparison to other approaches incorporating explicit temporal information for RL is missing (eg: LSTM based approaches).

########################

Questions during rebuttal:
- Please refer to questions in the Cons section and other feedback
- For results from Figure 6, why does Flare not outperform RAD for the hopper hop and walker run environments? Is there a way to visualize the temporal information to further investigate why Flare outperforms RAD in some environments (like quadruped walk) and not on others?

########################

Some typos and other feedback:
- Figure 2 and Section 5, paragraph 1, sentence 3: What is meant by proprioceptive state input? Consider formally defining it in the text for the reader.
- Section 5, paragraph 1, sentence 3: Consider ending the sentence by stating the fact that the experiments suggest the alternative to frame stacking heuristic is also effective in terms of performance.
- Consider defining p(a_t+1|a_t,o_t) as the transition function in the text for the reader.
- Section 5.1, last sentence: “… are done in the same except with augmented observations.” -> “… are done in the same way except with augmented observations.”
- Section 6, paragraph 1, sentence 1: “… that are experiments focus on.” -> “… that our experiments focus on.”
- Section 6.2, sentence 2:  You seem to have forgotten to mention the environments for which Flare outperforms RAD. What are you referring to with the phrase “remaining environments”?
- Section 6.2, sentence 5: “… walker run shown visualized in Figure 6.” -> “… walker run as shown in Figure 6.“
- Section 6.2, Figure 6: Why does Flare not perform as well as RAD for the Walker run environment?
- Conclusion, last sentence: Consider replacing “We would like to integrate Flare with model-based RL in the future” with “Integrating Flare with model-based RL is a potential direction for future work.”
- Consider replacing the usage of the phrase “state-based RL” with “RL on low-dimensional state space”.

---

> ### Author Response · Authors · 2020-11-22
> **Response to Reviewer 3**
>
> Thank you for spending time reviewing our paper and providing valuable feedback!
>
> We are glad that you find that the experiments are well-motivated and recognize that Flare could bring a broad impact. We have uploaded the revised manuscript to integrate your comments, where changes are highlighted in green.
>
> The specific responses to the reviews can be found below.
>
> Q1: Comparisons to recurrent baseline is missing
> A1: Thank you for suggesting this baseline. We agree with the reviewer that a comparison between Flare and recurrent architecture would further strengthen our analysis and highlight the advantages of Flare. Therefore, we have added results from a recurrent SAC to our state-based experiments in Section 4 with masked-out velocity information from the state-based input. Now figure 3 plots the results comparing Flare to (i) SAC with stacked consecutive states as inputs and (ii) recurrent SAC. Flare outperforms both stack SAC and recurrent SAC. More details can be found in the revised manuscript.
>
> Meanwhile, we would like to note that many well-developed state-of-the-art RL algorithms such as RAD and Rainbow DQN are non-recurrent. Although recurrent architecture is indeed another approach to explicitly aggregate temporal information, its additional overhead from training and implementation is significantly more costly than Flare.
>
> Q2: “Unclear why Flare works well on some but not others”
> A2:  It is very reasonable and common to have a new method of improving the majority of the environments but not all. For example, Rainbow DQN [1] delivers performance gain over Dueling DQN [2] on the majority of Atari games but not all. DrQ [3] outperforms Dreamer [4] on a subset of DMControl tasks while being comparable or even less competitive on the others.
>
> For the 6 additional environments in Appendix A.1, the baseline RAD [5] has optimally solved them. Learning from pixels with RAD is as efficient as from state SAC. For the sake of comprehensive benchmarking, we include these tasks despite that it is fair to say that gains on these tasks are saturated.
>
> Finally, we have also included results from a diverse subset of 8 Atari games in the revised manuscript with Flare applied to the Rainbow DQN baseline. Flare outperforms Rainbow on the majority (5 out of 8) while matching Rainbow on the remaining 3.
>
> Q3: Typos and other feedback
> A3: Again, thank you very much for your detailed feedback! Please refer to the revised manuscript where we have addressed the reviewer’s comments.
>
> [1] Reinforcement Learning with Augmented Data, Laskin et al Neurips 2020
>
> [2] Rainbow: Combining Improvements in Deep Reinforcement Learning, Hessel et al AAAI 2018
>
> [3] Dueling Network Architectures for Deep Reinforcement Learning, Wang et al ICML 2016
>
> [4] Image Augmentation Is All You Need: Regularizing Deep Reinforcement Learning from Pixels, Kostrikov et al 2020
>
> [5] Dream to control: learning behaviors by latent imagination, Hafner et al 2020 ICLR

---

### Official Review · AnonReviewer2 · 2020-10-28
**Frame stacking that is done on image embeddings**

**Rating:** 3
**Confidence:** 5

**Review:**

Significance:
The paper brings very little novelty or insight. It is unclear that the introduced architecture complexity worth marginal improvements (given high variance and only 5 random seeds) on 2 out of 11 tasks (5 from the main paper and 6 from appendix). This might be a good workshop paper but it clearly does not meet the high acceptance threshold of ICLR.

Pros:
-A simple architecture modification that might be beneficial to impose a stronger inductive bias on temporal dynamics.

Cons:
-The proposed algorithm is a trivial architecture change to the conv encoder, the introduced novelty is limited. Injecting the temporal difference inductive bias obviously will be beneficial, given that the sole reason for frame stacking is to infer velocity and acceleration.
-In Fig 3 the authors chose to only use 2 consecutive frames for State SAC, while a common practice is to use 3 frames. Using only 2 consecutive frames is not enough to infer acceleration and thus it is not a realistic setup, which makes this comparison meaningless. Also given that the variance is pretty high here, comparing performance over just 3 random seeds is not statistically conclusive.
-In Fig 6 the method is only evaluated only on 5 seeds and given that it demonstrates very high variance on the 3 tasks (out of 5) where it outperforms RAD it makes me think that the performance improvements are marginal and not worth introducing complexity.
-The ablation study is not very illuminating, this partially comes from the fact that the results are inconclusive (due to the high error bars), and partially because the experiments themselves are not very interesting.


Quality:
While the paper is well executed and made it significantly lacks on novelty and significance fronts.

Clarity:
The paper in general is clearly written and well organized.

---

> ### Author Response · Authors · 2020-11-22
> **Response to Reviewer 2 (Part 1)**
>
> Thank you for spending time reviewing our paper and providing valuable feedback.
>
> Q1: Unclear whether the introduced architecture complexity worth marginal improvements
> A1: We’d like to address the reviewer’s concerns over “architecture complexity” and “marginal improvements”  in the following:
>
> A.1.1 to Complexity overhead: The proposed algorithm Flare is in fact very simple to implement and can be easily integrated into many existing RL algorithms with a few lines of code modifications in the model file (see the attached model example in the supplementary materials). We answer in more detail in A2. Additionally, it does not add computational overhead.
>
> A.1.2 to Marginal Improvement: We argue that the empirical improvement from Flare is significant:
>
> --(i) the 5 core DMControl environments are **especially challenging**-- state-of-the-art model-free RL algorithms such as RAD [1] and CURL [6] are still far from state-based efficiency. On these challenging tasks, Flare substantially outperforms the state-of-the-art model-free baseline RAD [1] on the majority (3 out of 5),
>
> --(ii) for the 6 additional environments in Appendix A.1, the baseline RAD [1] has optimally solved them. Learning from pixels with RAD is as efficient as from state SAC. For the sake of comprehensive benchmarking, we include these tasks despite that it is fair to say that gains on these tasks are saturated.
>
> --(iii) we have also included results from a **diverse** subset of 8 Atari games in the revised manuscript with Flare applied to the Rainbow DQN baseline. **Flare outperforms Rainbow on the majority (5 out of 8)** while matching Rainbow on the remaining 3.
>
> --(iv) It is very reasonable and common to have a new method of improving a majority of the environments but not all. For example, Rainbow DQN [2] delivers performance gain over Dueling DQN [3] on the majority of Atari games but not all. DrQ’s [4] outperforms Dreamer [5] on a subset of DMControl tasks while being comparable or even less competitive on the others.
>
> For the reasons above, we argue that Flare substantially improves existing baselines with minimal changes, which we view as a positive and impactful contribution.
>
> Q2 Flare lacks novelty with minimal change and is trivial.
> We respectfully disagree with the reviewer over the characterization above.
>
> --On the one hand, Flare is indeed simple to implement, which we consider being a highly desirable property instead of a weakness. For one, its simplicity allows it to be generalized to many existing RL frameworks. For another, since Flare generally does not require additional hyperparameter tuning or increase model size, it takes very little overhead for practitioners to transfer it to their existing pipelines and to observe improvements. Therefore, we contend that Flare could have a broader impact.
>
> --On the other hand, although the high-level idea of incorporating temporal information is intuitive, effective execution of this idea is in fact very challenging and has not been done by prior work. In fact, the current deep RL community sticks to the legacy architecture of stacking frames in the input layer from Atari Nature-DQN (Mnih et al 2014), precisely because--in contrast to using optical flow in the computer vision community--it is **non-trivial** to incorporate explicit motion information in a computationally feasible and empirically effective fashion. We hope the reviewer reconsiders their judgment.
>
> --We reckon that positive contribution should not be solely measured by algorithmic complexity, for the most practically influential techniques often bear the property of easy to comprehend and simple to use, e.g. dropout, data augmentation, etc.

---

> ### Author Response · Authors · 2020-11-22
> **Response to Reviewer 2 (Part 2)**
>
> Q3: Given the high variance, results are inconclusive
> A2: Some environments intrinsically possess high variance due to various reasons, such as sparsity of reward. For example, in RAD [1] and CURL [6], Hopper Hop and Pendulum Swingup also display high variance. For another example, the baseline state SAC code we compare with [7], even though averaged over 10 seeds, still has large error bars in Hopper Hop, Pendulum Swingup, and Quadruped Walk.
>
> We would like to emphasize that our experiments use the same set of seeds to set the environments to compare different methods. Therefore, we don’t think the high variance could invalidate our results.
>
> Q3: Are 3/5 seeds enough?
> A3: Thank you for pointing this out. We agree that running multiple seeds is an important practice to demonstrate reliability and it is very common to run 3-5 seeds [5, 8, 9] on DMControl Suite in prior works. We assure the reviewers that we are doing our best to collect results on more seeds for both DMControl and Atari given the amount of compute resources we have access to.
>
> Q4: Using only 2 consecutive frames is not enough to infer acceleration
> A4: Thank you for pointing out this important detail being confusing. We have further clarified that 4 consecutive frames are used for state SAC in Fig 3 in the revision (see Fig 3 and Section 4).  This piece of information has in fact been reflected in the state Hyperparameter section in the Appendix.
>
> Q7: The ablation experiments are not interesting
> A7: Since Reviewer 1, 3, 4 find the ablation studies interesting and well-motivated, we hope to hear more specific feedback. We would be more than happy to add more ablations if it can provide the reviewer with more insight. We also added a state recurrent baseline in the revised manuscript that can be found in Section 4.
>
> [1] Reinforcement Learning with Augmented Data, Laskin et al Neurips 2020
>
> [2] Rainbow: Combining Improvements in Deep Reinforcement Learning, Hessel et al AAAI 2018
>
> [3] Dueling Network Architectures for Deep Reinforcement Learning, Wang et al ICML 2016
>
> [4] Image Augmentation Is All You Need: Regularizing Deep Reinforcement Learning from Pixels, Kostrikov et al 2020
>
> [5] Dream to control: learning behaviors by latent imagination, Hafner et al 2020 ICLR
>
> [6] CURL: Contrastive Unsupervised Representations for Reinforcement Learning, Srinivas et al 2020 ICML
>
> [7] https://github.com/denisyarats/pytorch_sac Yarats et al 2020
>
> [8] DeepMind Control Suite, Tassa, et al 2018
>
> [9] A Self-Tuning Actor-Critic Algorithm, Zahavy et al 2020 Neurips

---

### Official Review · AnonReviewer4 · 2020-10-28
**Official Blind Review #4**

**Rating:** 7
**Confidence:** 4

**Review:**

- Summary:
    - This paper presents Flare, an RL method that replaces frame stacking (early fusion) with latent vector stacking (late fusion) and then further improve upon this by adding in latent flow vectors (the difference between adjacent latent vectors)
    - The method is demonstrated on DM control using RAD-SAC as the baseline.  In all but one case, Flare outperforms the baseline
    - The authors then present a series of ablations to show that latent flow outperforms pixel flow and that stacking latent flow is better than pure latent stacking
- Strengths:
    - Relatively straightforward idea that improves upon the common technique of frame stacking, making this likely a technique with broad appeal and impact
    - Good ablation study showing how the individual components.
    - Well motivated idea.  I liked the experiment of position only SAC to set the motivation.
- Weaknesses
    - Concat + MLP seems like a poor way to do sequence modeling (a series of frames is, after all, a sequence).  How does Flare and the latent frame baseline behave if a GRU or LSTM is used to combine the the sequence of latent vectors?  This would avoid the normal challenges of training a recurrent policy while also benefiting from the superior sequence modeling of an RNN.
    - Performance on pendulum degrading as the number of frames is increased is concerning.  If possible, I wold like to see the hypothesis posed in Sec 6.3 Q3 validated by training longer.  Another possible hypothesis is that latent vector stacking increases the number of parameters and causes Q function overfitting.
    - Questions:
        - How was RAD applied to series of frames?  Was the same translation applied to all or was a different one applied too each?
- Overall
    - This paper presents and effective idea, however, there are some additional experiments (using an RNN to combine latent vectors) that I think would strength the paper considerably


## Post Rebuttal

I thank the authors for their response.  The addition of the recurrent SAC baseline helps the paper.  I disagree with R1 that it is a stronger baseline as FLARE outperforms it in all tasks and stack SAC similar or better three (arguably four) of five tasks. Instead it shows that recurrence isn't common in off-policy RL because it doesn't always perform better.  While recurrence is considerably more common in embodied 3D environments and this work may be less applicable there, I don't foresee DM control style RL benchmarks going away anytime soon and this believe this method will be useful.

---

> ### Author Response · Authors · 2020-11-23
> **Response to Reviewer 4**
>
> Thank you for spending time reviewing our paper and providing valuable feedback. We especially appreciate your patience as we gather experimental results to address the questions raised in the review.
>
> We are glad that you find the experiments well-motivated and find the technique could have a broad impact. We respond to the reviews in detail below.
>
> Q1: “How do Flare and the latent frame baseline behave if a GRU or LSTM is used to combine the sequence of latent vectors?”
>
> A1: Thank you for suggesting this baseline. We agree with the reviewer that a comparison between Flare and recurrent architecture would further strengthen our analysis and highlight the advantages of Flare. Therefore, we have added results from a recurrent SAC to our state-based experiments in Section 4 with masked-out velocity information from the state-based input. Now figure 3 plots the results comparing Flare to (i) SAC with stacked consecutive states as inputs and (ii) recurrent SAC. **Flare outperforms both stack SAC and recurrent SAC.** More details can be found in the revised manuscript.
>
> Meanwhile, we would like to note that many well-developed state-of-the-art RL algorithms such as RAD and Rainbow DQN are non-recurrent. Although recurrent architecture is indeed another approach to explicitly aggregate temporal information, its additional overhead from training and implementation is significantly more costly than Flare.
>
> Q2: How was RAD applied to a series of frames? Was the same translation applied to all or was a different one applied to each?”
>
> A2: RAD [1] applies the same translation transformation to the stack of three frames per data point. We follow the same protocol in our experiments.
>
> Q3: Pendulum Swingup performance goes down while increasing the number of frames.
>
> A3: In Section 6.3 Q3, we hypothesize that Flare trains more slowly with increased frame count on Pendulum Swingup due to the presence of unnecessary information that the actor and critic networks need to learn to ignore.
>
> Following the reviewer’s suggestion, we rerun Flare with 2, 3, and 5 frames but train longer over 3 seeds (due to time constraint, some of the runs are still in progress) and present the results on each seed individually and observe:
>  (link to the plot: https://drive.google.com/file/d/1lumTJ0l7IiDw8ssUATU5rA1zcTMmyg0q/view?usp=sharing)
>
> (i) indeed, Flare converges slower with a higher input frame count, which agrees with our hypothesis.
>
> (ii) the 3rd seed for 5-frame Flare fails to converge, which explains the poor performance of 5-frame Flare from Figure 8 in the main text.
>
>
> We thank the reviewer for suggesting the overfitting hypothesis, but carefully contemplate that it is unlikely the case for the following reasons:
> (i) we supply a generous amount of data augmentation during training.
>
> (ii) we don’t observe similar degradation on quadruped walk hence it is more likely linked to the intrinsic properties of the Pendulum Swing environment.
>
> (iii) we bottleneck the concatenated feature $(z_1, z_2 .. z_k, \delta_1, \delta_2, .. \delta_k)$ to a very low dimension latent space (50 hidden units) before advancing to the actor and critic networks. Note that the same architecture is used by the baseline RAD-SAC. The details can be found in the code, `pixel_flare/encoder.py`  L123-143, and also pasted below:
> ```python
> def forward(self, obs, detach=False, delta=False):
>       h = self.forward_conv(obs)
>       if detach:
>           h = h.detach()
>
>       try:
>           h_fc = self.fc(h)
>       except:
>           print(obs.shape)
>           print(h.shape)
>           assert False
>       self.outputs['fc'] = h_fc
>
>       h_norm = self.ln(h_fc)
>       self.outputs['ln'] = h_norm
>       if self.output_logits:
>           out = h_norm
>       else:
>           out = torch.tanh(h_norm)
>           self.outputs['tanh'] = out
>       return out
> ```
> [1] Reinforcement Learning with Augmented Data, Laskin et al Neurips 2020

---

### Official Review · AnonReviewer1 · 2020-10-29
**Review 1**

**Rating:** 4
**Confidence:** 3

**Review:**

This paper presents a new method for aggregating temporal information in reinforcement learning policies. The method takes the difference of latent representations between consecutive frames and concatenates this difference with the latent representations for downstream processing.

Strengths:
- The method is simple and easy to understand and implement.
- The method outperforms a baseline on average based on experiments on 5 Deepmind control benchmark suite tasks under a limited budget of training samples.
- Ablations show an interesting result that the fusion of latent representations performs better than pixel-based fusion.

Weaknesses:
- The contribution is incremental in my opinion. It makes a small change over existing RL methods. The improvement in performance on several tasks is also marginal.
- One of the most common methods of aggregating temporal information is just using a recurrent layer in RL policies. It’s unclear if the baseline consists of a recurrent layer. If it does not, a simple recurrent policy needs to be added as a baseline. If it does, it is unclear to me why a simple subtraction of latent representations would result in performance improvement. I would imagine that it should be very easy for a recurrent layer such as an LSTM to learn differences between latent representations if they were useful. Some discussion on this would be useful.
- The performance gain is only demonstrated on 5 tasks and it’s unclear whether it would translate to gains in other tasks as well. Results in the appendix on 6 easier tasks show results comparable to or worse than the baseline.

Overall, I believe that the paper proposes a simple and interesting method, but I am not convinced that it would lead to better results consistently. The main experiments are conducted only on 5 tasks and the performance gains are marginal in my opinion.

Update after rebuttal:
The authors have added a recurrent SAC baseline to one set of experiments. The results indicate that the recurrent SAC is a much stronger baseline, and the variance of results is high enough that I am not convinced of the benefits of the proposed method.

The authors argue that "many state-of-the-art RL algorithms" "are non-recurrent", and "frame stacking" is "largely untouched since its inception and is used in most state-of-the-art RL architectures", "recurrent architectures" have "additional overhead from training and implementation". I do not believe this is true. Recurrent architectures are commonly used in RL algorithms (for example RSSM in Dreamer) and are widely available in open-source implementations (for example https://github.com/openai/baselines/blob/master/baselines/common/models.py). There are some prior papers which use the frame stacking heuristic for a fair comparison with DQN, but this heuristic or the non-recurrent model architecture is not a part of the RL algorithm itself. Since this paper proposes a method for extracting temporal information, LSTM/GRU are very natural baselines in my opinion and should be added to all experiments.

The authors argue that "it is very reasonable and common to have a new method improving a majority of the environments but not all", I agree with this, but the examples given by the authors such as Rainbow DQN, Dueling DQN, Dreamer etc perform experiments in many more environments and performance improvements are larger. I believe a much larger scale study is needed to compare Flare with recurrent baselines and make conclusive statements about performance gains.

---

> ### Author Response · Authors · 2020-11-22
> **Response to Reviewer 1**
>
> Thank you for spending time reviewing our paper and providing valuable feedback.
>
> We are glad that you find the proposed method well-explained and the ablation studies interesting. Given that the main concerns are around i) marginal improvement ii) Flare’s transferability to other tasks iii) lacking a recurrent baseline, we have provided additional results on Atari games and added a recurrent baseline. The following addresses your comments in detail.
>
> Q1: “small change” with “marginal improvement”
> A1:  We’d like to address the reviewer’s concerns over “marginal improvement” and “small change”  in the following:
>
> A1.1 to “marginal improvement”,  “unclear whether this would transfer” or “lead to better results consistently”: We argue that the empirical improvement from Flare is significant, transferable and consistent in the following:
>
> --(i) the 5 core DMControl environments are **especially challenging** and Flare substantially outperforms the baselines on the majority (3 out of 5),
>
> --(ii) for the 6 additional environments in the Appendix A.1, the baseline RAD [1] has optimally solved them. Learning from pixels with RAD is as efficient as from state SAC. For the sake of comprehensive benchmarking, we include these tasks despite that it is fair to say that gains on these tasks are saturated.
>
> --(iii) To demonstrate how Flare transfers to other tasks and RL algorithms, we have also included results from a **diverse** subset of 8 Atari games in the revised manuscript with Flare applied to the Rainbow DQN baseline. Flare **outperforms Rainbow on the majority (5 out of 8)** while matching Rainbow on the remaining 3.
>
> --(iv) It is very reasonable and common to have a new method improving a majority of the environments but not all. For example, Rainbow DQN [2] delivers performance gain over Dueling DQN [3] on the majority of Atari games but not all. DrQ’s [4] outperforms Dreamer [5] on a subset of DMControl tasks, while being comparable or even less competitive on the others.
>
> For the reasons above, we argue that Flare substantially improves existing baselines with minimal changes, which we view as a positive and impactful contribution.
>
> A1.2 to “small change”:
> --On the one hand, Flare is indeed simple to implement, which we consider to be a highly desirable property instead of a weakness. For one, its simplicity allows it to be generalized to many existing RL frameworks. For another, since Flare generally does not require additional hyperparameter tuning or increase model size, it takes very little overhead for practitioners to transfer it to their existing pipelines and to observe improvements. Therefore, we contend that Flare could have a broader impact.
>
> --On the other hand, although the high-level idea of incorporating temporal information is intuitive, an effective execution of this idea is in fact very challenging and has not been done by prior work. In fact, current deep RL community sticks to the legacy architecture of stacking frames in the input layer from Atari Nature-DQN (Mnih et al 2014), precisely because--in contrast to using optical flow in computer vision community--it is **non-trivial** to incorporate explicit motion information in a computationally feasible and empirically effective fashion. We hope the reviewer reconsiders their judgement.
>
> --We reckon that positive contribution should not be solely measured by algorithmic complexity, for the most practically influential techniques often bear the property of easy to comprehend and simple to use, e.g. dropout, data augmentation, etc.
>
> Q3: recurrent baseline:
> A3: Thank you for suggesting this baseline. We agree with the reviewer that a comparison between Flare and recurrent architecture would further strengthen our analysis and highlight the advantages of Flare. Therefore, we have added results from a recurrent SAC to our state-based experiments in Section 4 with masked-out velocity information from the state-based input. Now figure 3 plots the results comparing Flare to (i) SAC with stacked consecutive states as inputs and (ii) recurrent SAC. **Flare outperforms both stack SAC and recurrent SAC.** More details can be found in the revised manuscript.
>
> Meanwhile, we would like to note that many well-developed state-of-the-art RL algorithms such as RAD and Rainbow DQN are non-recurrent. Although recurrent architecture is indeed another way to explicitly aggregate temporal information, its additional overhead from training and implementation is significantly more costly than Flare.
>
> [1] Reinforcement Learning with Augmented Data, Laskin et al Neurips 2020
>
> [2] Rainbow: Combining Improvements in Deep Reinforcement Learning, Hessel et al AAAI 2018
>
> [3] Dueling Network Architectures for Deep Reinforcement Learning, Wang et al ICML 2016
>
> [4] Image Augmentation Is All You Need: Regularizing Deep Reinforcement Learning from Pixels, Kostrikov et al 2020
>
> [5] Dream to control: learning behaviors by latent imagination, Hafner et al 2020 ICLR

---

### Author Response · Authors · 2020-11-20
**Revision Uploaded**

Dear Reviewers,

Thank you very much for your valuable feedback and time spent reviewing our paper! We have updated the main text manuscript and the supplementary materials to reflect your comments. The revised texts are highlighted in **green**.

The major updates are summarized as follows:
-- we have implemented recurrent SAC and compared it to the proposed Flare. Flare outperforms recurrent SAC.
-- we have incorporated Flare to Rainbow DQN and compared them on a subset of challenging Atari games. Flare outperforms Rainbow on the majority (5 out of 8) and matches on the remaining ones. Code is attached in the supplementary materials.
-- we have clarified certain important details which the reviewers have kindly pointed out.

We will also address your feedback individually in the next few days.

Thank you very much again!

Best,
anonymous authors

---

### Decision · Program_Chairs · 2021-01-07
**Final Decision**

**Decision:**

Reject

**Comment:**

This paper provides a simple approach to incorporate temporal information in RL algorithms. AC agrees with authors that simplicity is a virtue. As reviewers point out that experimentally the approach is not conclusively better (given that environments might be hand-chosen). Even R3 believes some reported improvements is within variance. Given the discussions, AC agrees that results do not seem convincing enough.